# Combined Inhibition of TGF-β Signaling and the PD-L1 Immune Checkpoint Is Differentially Effective in Tumor Models

**DOI:** 10.3390/cells8040320

**Published:** 2019-04-05

**Authors:** Heng Sheng Sow, Jiang Ren, Marcel Camps, Ferry Ossendorp, Peter ten Dijke

**Affiliations:** 1Department of Cell and Chemical Biology, Oncode Institute, Leiden University Medical Center, 2300 RC Leiden, The Netherlands; h.s.sow@lumc.nl (H.S.S.); j.ren@lumc.nl (J.R.); 2Department of Immunohematology and Blood Transfusion, Leiden University Medical Center, 2300 RC Leiden, The Netherlands; m.g.m.camps@lumc.nl

**Keywords:** anti-PD-L1 mAb 1, LY364947 2, mouse syngeneic tumor models 3

## Abstract

Antibodies blocking the programmed death-ligand 1 (PD-L1) have shown impressive and durable responses in clinical studies. However, this type of immunotherapy is only effective in a subset of patients and not sufficient for rejection of all tumor types. In this study, we explored in two mouse tumor models whether the antitumor effect could be enhanced by the combined blockade of PD-L1 and transforming growth factor-β (TGF-β), a potent immunosuppressive cytokine. The effect of anti-PD-L1 mouse monoclonal (mAb) and a TGF-β type I receptor small molecule kinase inhibitor (LY364947) was evaluated in the highly immunogenic mouse MC38 colon adenocarcinoma and the poorly immunogenic mouse KPC1 pancreatic tumor model. In the MC38 tumor model, LY364947 monotherapy did not show any antitumor effect, whereas treatment with anti-PD-L1 mAb significantly delayed tumor outgrowth. However, combination therapy showed the strongest therapeutic efficacy, resulting in improved long-term survival compared with anti-PD-L1 mAb monotherapy. This improved survival was associated with an increased influx of CD8^+^ T cells in the tumor microenvironment. In the KPC1 tumor model, LY364947 did not enhance the antitumor effect of anti-PD-L1 mAb. Despite this, delayed KPC1 tumor outgrowth was observed in the LY364947-treated group and this treatment led to a significant reduction of CD4^+^ T cells in the tumor microenvironment. Together, our data indicate that an additive anti-tumor response of dual targeting PD-L1 and TGF-β is dependent on the tumor model used, highlighting the importance of selecting appropriate cancer types, using in-depth analysis of the tumor microenvironment, which can benefit from combinatorial immunotherapy regimens.

## 1. Introduction

Immune checkpoint molecules are gaining prominence as targets for cancer immunotherapy, demonstrating durable remission of patients with metastatic lesions [1]. Last year the Nobel prize for physiology and medicine was awarded to James P. Allison and Tasuku Honjo “for their discovery of cancer therapy by inhibition of negative immune regulation” [2]. Antibodies targeting programmed death ligand 1 (PD-L1) such as atezolizumab, avelumab, and durvalumab have received regulatory approval [3,4,5,6]. Despite showing remarkable durable remissions, these antibodies only demonstrate their efficacy in a subset of specific cancer types [7]. In order to increase the therapeutic efficacy, many on-going preclinical and clinical studies are evaluating anti-PD-L1 mAb in combination with other immunostimulatory agents or cancer-modulating drugs. An important strategy is to down-regulate the immune suppression that is elicited by the tumor microenvironment to allow immunotherapy to be effective.

Transforming growth factor-β (TGF-β) is an immunosuppressive cytokine which is often produced in large quantities by many cell types in the tumor microenvironment, including tumor cells [8,9]; regulatory T cells [10,11]; and myeloid suppressor cells [12,13]. TGF-β is well known for its pleiotropic role from initiating to promoting tumor development [14,15,16,17] and it has a negative effect on anti-tumor immunity by suppressing the effector functions of several immune effector cells such as neutrophils, macrophages, natural killer (NK) cells, CD8 cells, and CD4 T cells [16,18,19,20]. Together with other cytokines such as interleukin (IL)-2 and IL-6, TGF-β also induces the generation and recruitment of regulatory T cells to further suppress the antitumor T and NK cell responses [21,22]. Moreover, it is also known for its role in regulating and promoting the accumulation of stiff fibrillary extracellular matrix composed of collagen [23], resulting in hindered drug transport [24] and infiltration of immune cells [25,26,27] into the tumor. Most importantly, high serum levels of three TGF-β isoforms, TGF-β1, TGF-β2, and TGF-β3, correlate with poor clinical outcome [28,29,30,31,32].

As such, it is plausible that TGF-β inhibition, through reducing immune suppression and decreasing deposition of matrix collagen content, could potentially improve infiltration of activated immune effector cells and delivery of drug into the tumor microenvironment. In this study, we investigated if the treatment of TGF-β receptor 1 selective small molecule kinase inhibitor, termed LY364947 [33], can enhance the antitumor efficacy of anti-PD-L1 mAb in immunogenic (MC38 colorectal tumor) and poorly immunogenic (KPC1 pancreatic tumor) tumor models. 

## 2. Materials and Methods

### 2.1. Cell Culture

The mouse breast cancer TUBO cell line was a gift from Prof Guido Forni [34]; mouse pancreatic tumor cell lines KPC1 and KPC3 were obtained from Dr Thorsten Hagemann (Queen Mary, University of London). B16OVA, a variant of the melanoma B16F10 tumor line that expresses full-length OVA, was a gift from K. L. Rock (University of Massachusetts Medical Center, Worcester, MA, USA). EL4 and B16F10 cell lines were obtained from ATCC (Rockville, MD, USA). RMA is a mutagenized derivative of RBL-5, a Rauscher MuLV-induced T lymphoma cell line [35]. The MC38 tumor cell line is derived from a primary mouse colon carcinoma [36]. The C3 tumor cell line was generated by transfection of B6 mouse embryonic cells (MEC) with the complete HPV16 genome and maintained as previously described [37]. All tumor cell lines were cultured in Iscove’s modified Dulbecco’s medium (IMDM) (Lonza, Allendale, NJ, USA) supplemented with 10% heat-inactivated fetal bovine serum (FBS) (Greiner, Bio-One, Frickenhausen, Germany), 2 mM l-glutamine (Gibco, Invitrogen, Blijswijk, The Netherlands), 25 µM 2-mercaptoethanol (Merck, Darmstadt, Germany), and 100 IU/mL penicillin/streptomycin (Gibco). Dulbecco’s modified Eagle’s medium (DMEM) supplemented with 10% heat-inactivated FBS (Greiner, Bio-One, Frickenhausen, Germany) and 100 U/mL penicillin/streptomycin (Gibco) were used to culture human embryonic kidney (HEK)293 cells. HEK293 cells were obtained from ATCC (Rockville, MD, USA). All cell lines in our studies were maintained at 37 °C, with 5% CO_2_, in a humidified incubator and were free of mycoplasma. 

### 2.2. Mice

Wild-type (WT) C57Bl/6 female mice were purchased from Charles River (L’Arbresle, France) and maintained under specific pathogen-free (SPF) animal facilities of the Central Animal Facility (PDC) of the Leiden University Medical Center (LUMC). Mice were 8–9 weeks old at the beginning of each experiment. The health status of the animals was monitored over time. Animals tested negative for all agents listed in the Federation of European Laboratory Animal Science Associations (FELASA) guidelines for SPF mouse colonies [38]. All animal studies were approved by the animal ethics committee of LUMC. Experiments were performed recommendations and guidelines set by LUMC and the Dutch Act on Animal Experimentation and EU Directive 2010/63/EU (Guidelines on the Protection of Experimental Animals).

### 2.3. Syngeneic Tumor Studies

MC38 colon adenocarcinoma cancer cells (4 × 10^5^ cells) were injected subcutaneously into 8–12-week-old mice in 100 µL of phosphate buffered saline (PBS). Then, 200 µg of anti-PD-L1 mAb (clone MIH5) were injected intraperitoneally at days 6, 8, and 11 after inoculation. LY364947 was purchased from Selleckchem (Huston, TX, USA) and dissolved in dimethylsulfoxide (DMSO) to make final concentration of 20 mg/mL. Then, 10 mg/kg of LY364947 were injected intraperitoneally at days 6, 8, and 11 and once every three days after cancer cell inoculation. The KPC1 pancreatic cancer cell line was generated from *Kras^LSL-G12D/+^*, *Trp53^LSL-R172H/+^*, *Pdx1-Cre* (KPC) mice and was a gift from Thorsten Hagemann (Queen Mary University of London). The tumor cells (1 × 10^5^ cells) were injected subcutaneously into 8–12-week-old mice in 100 µL of PBS. At days 9, 11, and 14 post tumor inoculation, mice were injected intraperitoneally with 200 µg of anti-PD-L1 mAb (clone MIH5). For the LY364947 or combination group, mice received 10 mg/kg of LY364947 (intraperitoneally) at day 9 and once every day post tumor inoculation. All tumors were measured twice weekly using calipers. Mice were sacrificed when tumors reached a size of 100 mm^2^ to avoid unnecessary suffering. Both cell lines were mycoplasma and mouse antibody production (MAP)-tested before the start of tumor studies.

### 2.4. Flow Cytometry

Harvested tumors were manually minced into small pieces with scalpels before incubating with 350 μg/mL Liberase TL (Roche) for 20 min at 37 °C and filtered through a 70-µm cell strainers (BD Biosciences, Bedford, MA, USA) to obtain single cell suspension. The cells were subjected to Ammonium-Chloride-Potassium (ACK) lysis (5 min) before staining with 10% normal mouse serum and anti-mouse CD16/CD32 antibody (clone 2.4G2) to block Fc receptor for IgG (FcγRs). Single-cell suspensions of tumor-infiltrating lymphocytes were stained using the following antibodies: CD8α (clone 53-6.7), CD4 (clone L3T4), CD3ε (clone 145-2c11), CD11b (clone M1/70), F4/80 (clone BM8), CD45.2 (clone 104), Ly6G (clone 1A8), PD-L1 (clone MIH5). LAG-3 (C9B7W), and CTLA-4 (9H10). Then, 7-AAD staining (Invitrogen, Carlsbad, CA, USA) was used to exclude dead cells. All stained cells were analyzed on a LSRII cytometer (BD) and data analysis was performed with FlowJo Software v10 (Tree Star, San Carlos, CA, USA).

### 2.5. mTGF-β1 ELISA

Briefly, tumor cell lines were cultured in 24-well plates in complete IMDM until 80% confluent. Cells were washed twice with PBS and cultured in IMDM supplemented with 1% FBS (not heat-inactivated) for 24 h at 37 °C. Supernatants were collected and stored at −20 °C until further analysis. Total mTGF-β1 levels were measured by using a Mouse TGF-β1 duoset ELISA kit according to the manufacturer’s instructions (#DY1679, R&D Systems, Minneapolis, MN, USA).

### 2.6. CAGA Luciferase Reporter Assay

To produce conditional medium (CM), MC38, KPC1, KPC3, and B16F10 cells were washed two times with PBS at 70–80% confluency and incubated in serum-free DMEM medium for 24 h. CM was then collected and passed through a 0.45-mm Syringe Filter (SLHP033RB, Merck Millipore, Billerica, MA, USA). HEK293 cells were seeded at approximately 5 × 10^4^ cells per well into a 24-well plate. The next day, cells in each well were co-transfected with 0.1 µg TGF-β/SMADinducible (CAGA)_12_ luciferase transcriptional reporter construct, which encodes 12 repeats of the AGCCAGACA sequence (identified as a SMAD3/SMAD4-binding element in the human *PAI-1* promoter [39]), and 0.08 µg β-galactosidase construct (driven by a cytomegalovirus promoter) using five times of polyethyleneimine in quantity. After overnight incubation, HEK293 cells were starved with serum free medium. Eight hours later, serum free media were removed and replaced by CM. A TGF-β treatment (5 ng/mL, 8420-B3, R&D SYSTEMS, Minneapolis, MN, USA) was also performed that served as a standard. After another overnight incubation, luciferase and β-galactosidase activities were measured. The luciferase activity was normalized based on the β-galactosidase activity. Representative experiments indicating the mean and standard deviation of triplicate values are shown.

### 2.7. Western Blot

Approximately 2.5 × 10^5^ of MC38 and KPC1 cells were plated in 6-well plate in complete medium and incubated overnight at 37 °C. The next day, the complete medium was replaced with 0.2% FBS medium and further incubated at 37 °C for eight hours. Cells were then treated with 1 µg/mL of LY364947 for 30 min before stimulating with 5 ng/mL of TGF-β3 for 2 h. Cells were lysed in radioimmunoprecipitation assay buffer (RIPA) sampler buffer (50 mM Tris–HCl (pH 8.0) with 150 mM NaCl, 1.0% Nonidet P-40, 0.5% sodium deoxycholate, and 0.1% sodium dodecyl sulfate) containing cOmplete™ Protease Inhibitor Cocktail (11697498001, Roche, Basel, Switzerland). Protein concentration was determined using a DC™ Protein Assay Kit (5000111, Bio-Rad, Hercules, CA, USA). An equal amount of protein was subjected to sodium dodecyl sulfate–polyacrylamide gel electrophoresis and blotted onto a polyvinylidene difluoride membrane (IPVH00010, Merck Millipore). Membrane was probed with phospho-SMAD2 antibody [40] (homemade) and GAPDH antibody (AB2302, Merck Millipore, Billerica, MA, USA). The chemiluminescent signal was detected using the Clarity™ Western ECL Substrate (Hercules, CA, USA) and visualized using the ChemiDoc™ Imaging Systems (17001402, Bio-Rad, Hercules, CA, USA).

### 2.8. In Vitro Cell Proliferation Assay

MC38 and KPC1 cells were plated in a 96-well plate, with 2 × 10^3^ cells/well approximately, and incubated overnight. Cells were treated with vehicle control, or 1 µg/mL of LY364947, or 5 ng/mL of TGF-β3, or a LY364947 and TGF-β3 combination. The cell proliferation was determined by CellTiter 96^®^ AQueous Non-Radioactive Cell Proliferation Assay (MTS) (G5421, Promega BioSciences, Madison, WI, USA) following the manufacturer’s protocol. Absorbance was measured at 490 nm over 5 consecutive days using VICTORX Multilabel Plate Reader (2030-0050, Perkin Elmer, Waltham, MA, USA). Each group was evaluated in five repeats, and a cell growth curve was plotted.

### 2.9. Statistical Analyses

Data were analyzed using Prism 7.0 GraphPad Prism 7.0 (GraphPad Software, La Jolla, CA, USA). To determine statistical significance between two groups, an unpaired Student’s *t*-test was performed. Significance between more than two groups was evaluated by one-way ANOVA. Kaplan–Meier method and the log-rank (Mantel–Cox) test were used to determine statistical differences in the survival of mice.

## 3. Results

### 3.1. Colorectal and Pancreatic Cancer Cells Produce High Levels of mTGF-β1

In order to select mouse tumor models to investigate the therapeutic efficacy of combining TGF-β inhibitor and anti-PD-L1 mAb, we measured mTGF-β1 production by various mouse tumor cell lines. As illustrated in Figure 1A, ELISA analysis revealed that both pancreatic (KPC1) and colorectal (MC38) cancer cell lines produced high levels of latent mTGF-β1 protein. Using a transcriptional reporter assay, we observed that MC38 but not KPC1 cells secreted elevated amounts of active mTGF-β (Figure 1B). Due to the high level of production of latent and/or active mTGF-β, MC38 and KPC1 were selected for in vivo analysis. We first evaluated the TGF-β/SMAD2 response and efficacy of the small molecule inhibitor LY364947 targeting the TGF-βRI serine/threonine kinase activity in both cell lines when cultured in vitro. TGF-β potently stimulated the phosphorylation of SMAD2 (pSMAD2) in MC38 and KPC1 cell lines and this was blocked by LY364947 (Figure 1C). Despite these inhibitory effects, the proliferation of tumor cells remained unaffected by LY36947 and/or TGF-β treatment (Figure 1D). Next, the effect of LY364947 treatment in vivo was determined by investigating intra-tumoral levels of pSMAD2 after intraperitoneal (i.p.) injection of LY364947 in mice bearing either established MC38 or KPC1 tumors (Figure 1D). Histology analysis using phospho-SMAD2 antibody revealed strong phosphorylation of SMAD2 in control DMSO and 1 and 4 h post LY364947-treated MC38 and KPC1 tumors. Decreased TGF-β-induced SMAD2 phosphorylation was observed in 8 h post LY364947-treated tumors and this inhibitory effect of LY364947 appeared to last longer in MC38 than KPC1 tumors. 

### 3.2. TGF-β Kinase Inhibitor LY364947 Improves Therapeutic Efficacy of Anti-PDL1 mAb

The MC38 colon adenocarcinoma syngeneic model on a C57BL/6 background is highly immunogenic and it has been demonstrated to be sensitive to anti-PD-L1 immune checkpoint monotherapy [41,42]. To test if LY364947 boosts the antitumor effect of anti-PD-L1 mAb, we examined the anti-tumor effect of these treatments on subcutaneously growing MC38 tumors in immune-competent C57BL/6 mice. As shown in Figure 2A, LY364947 induced little therapeutic effect, whereas treatment with anti-PDL1 mAb or combination therapy significantly delayed tumor outgrowth, leading to prolonged overall survival. Beyond day 31, the survival rate of mice treated with combination therapy showed significantly higher survival rate than mice receiving anti-PD-L1 mAb (Figure 2A). These data suggest that the blockade of TGF-β receptor activity enhanced the anti-tumor immunity of anti-PD-L1 mAb therapy, leading to improved overall long-term survival in the immunogenic MC38 tumor model (Figure 2B). 

### 3.3. Effect of Combination Therapy on the MC38 Tumor Microenvironment

To investigate the mechanism of action of anti-PDL1 mAb and LY364947 in the MC38 tumor model, we first analyzed the impact of the therapies on the frequency of immune cells in the tumor microenvironment of MC38 tumors by flow cytometry. As shown in Figure 3A, treatment with combined therapy of LY364947 and anti-PDL1 mAb led to a higher frequency of tumor-infiltrating CD3^+^ T cells. CD8^+^ (Figure 3B, Appendix A) but not CD4^+^ (Figure 3C). T cells were accountable for the higher frequency to tumor-infiltrating T cells. Moreover, blockade of TGF-β had no effect on the frequency of Foxp3^+^ CD4^+^ T cells (Appendix A). Frequencies of F4/80^+^ macrophages (Figure 3D) and Ly6G^+^ granulocytes (Figure 3E) were not significantly affected by LY364947 and combination therapy. Our results support previously reported studies which show that the combination of TGF-β and PD-L1 blockade increased the percentages of CD8^+^ T effector cells in the tumor bed [27] which correlates with the improved tumor eradication of the combination treatment.

### 3.4. TGF-β Inhibitor Delays KPC1 Pancreatic Tumor Outgrowth

Unlike the MC38 colorectal tumor model which is known to have high mutational load [43], the KPC tumors, which are derived from the KPC transgenic mouse strain which drives pancreatic ductal adenocarcinoma (PDA) tumorigenesis by expression of a combination of strong oncogenes, is a poorly immunogenic tumor due to a low mutational burden [44]. To investigate the potential checkpoint inhibitors in this PDA model, we examined the expression of programmed cell death protein 1 (PD-1), cytotoxic T-lymphocyte-associated protein 4 (CTLA-4), and lymphocyte-activation gene 3 (Lag-3) on T cells within the KPC tumor microenvironment. In KPC tumor, the infiltrating T cells were predominantly CD4^+^ and majority of them expressed PD-1 (Figure 4A). These data suggest that blockade of PD-1/PD-L1 may improve antitumor immunity in KPC1 tumor model. 

We therefore tested the effect of anti-PD-L1 mAb and LY364947 on KPC1 pancreatic tumor outgrowth. Treatment with anti-PD-L1 mAb did not impact tumor outgrowth. In contrast, treatment with LY364947 or combination therapy significantly reduced tumor outgrowth as compared to untreated group (Figure 4B). This suggest that anti-tumor effect was most likely elicited by blocking of TGF-β signaling pathway. Moreover, flow cytometric analysis revealed a decrease of total CD3^+^ T cells, particularly CD4^+^ T cells (Figure 4C; Appendix A), but no detectable decrease of Foxp3^+^ CD4^+^ T cells (Appendix A). No reduction was observed of granulocytes and macrophages (Figure 4C).

## 4. Discussion

Experimental tumor models are essential preclinical step for the development and evaluation of cancer immunotherapy strategies. From our studies with the MC38 and KPC1 tumor models, one key finding that emerged is that tumor immunogenicity is a dominant feature predicting responsiveness to dual targeting of TGF-β signaling and PD-L1. In an immunogenic MC38 tumor model, blocking PD-L1 significantly delayed MC38 tumor outgrowth. However, combination LY364947 with anti-PD-L1 mAb further improved overall survival versus anti-PD-L1 mAb monotherapy. The antitumor activity of this combination therapy is consistent with the findings of multiple recent studies using immunogenic tumor models which demonstrated the improvement of anti-PD-L1 mAb when it is combined TGF-β receptor kinase inhibitor galunisertib [45,46]. In all studies investigating the therapeutic efficacy of galunisertib in the colon adenocarcinoma model, galunisertib was injected at high amounts (from 75 mg/kg to 800 mg/kg) and frequent intervals. This might explain the limited antitumor effect of LY364947 (10 mg/kg) monotherapy on MC38 tumor outgrowth observed in our study. Nonetheless, we show that the anti-tumor activity of the combination therapy is associated with higher levels of tumor-infiltrating CD8^+^ T cells. This observation is in agreement with the finding of Mariathasan et al. [27] who demonstrated that the main mechanism of action of TGF-β is to increase T-cell infiltration into MC38 tumor. Together, these data suggest that co-administration of TGF-β and PD-L1 blocking agents may provide a subset of colorectal cancer patients a more favorable outcome.

On the other hand, a combined effect of anti-PD-L1 mAb and LY364947 was not observed in poorly immunogenic KPC1 tumor model; blocking of TGF-β resulted in significant reduction of KPC1 tumor outgrowth, in contrast to the anti-PD-L1 mAb-treated group, which was not effective in this model. This lack of antitumor efficacy is similar to the lack of responses observed in KPC tumor bearing animal treated with anti-PD1 and/or anti-CTLA-4 mAb [47]. The limited effect of immune checkpoint inhibitors in this tumor model may be due to the low mutational burden and absence of potential neoepitopes derived from tumor mutations [44]. This model is reminiscent of most human pancreatic cancers with similar low numbers of mutations [48]. For this reason, the KPC pancreatic tumor model has a high potential of translational relevance for examining therapeutic efficacy of anti-PD-L1 mAb and LY364947. However, a small cohort of pancreatic cancer patients has been shown to have a relatively high mutational burden [49,50]; this may have an impact on the therapeutic efficacy of anti-PD-L1 mAb and LY364947. Therefore, study with an alternative pancreatic tumor cell line such as Pan02 (derived from Pancreatic ductal adenocarcinoma (PDAC) tumor induced by implanting 3-methyl-cholanthrene in the pancreas of C57Bl/6 mice) [51] that has a higher mutational burden may help address this question.

More evidence is emerging that targeting TGF-β can elicit beneficial effects in halting the pancreatic tumorigenic process. In a study by Principe and colleagues [52], global loss of TGF-β signaling protected against pancreatic tumor development via inhibition of tumor-associated fibrosis, stromal TGF-β1 production, and restoration of anti-tumor CD8^+^ T cells responses. Here we showed that the treatment with LY364947 independent of the established subcutaneous KPC1 tumor decreases the relative amount of CD4^+^ T cells within the tumor microenvironment. The potential role of CD4^+^ T cells in promoting pancreatic tumorigenesis has been reported by Alam et al. who showed that the p38 MAP kinase inhibitor induced a reduction in the percentage of CD4^+^ tumor infiltrating lymphocytes (TILs) producing tumor necrosis factor (TNF)-α, retinoic acid-related orphan receptor (RORγt), interferon γ (IFNγ), and interleukin (IL)-17, and was associated with improved survival in KPC tumor-bearing animals. A significant delay in pancreatic intraepithelial neoplasia (PanIN) was reported in spontaneous pancreatic tumor model KC mice that received weekly CD4-depleting antibodies [53]. Although TGF-β might also be expected to reduce the regulatory CD4^+^ T cells (Tregs) cell population [11,54], our data suggest that the numbers of Tregs are not strongly affected by LY364947 and therefore future investigations are warranted to reveal the subsets of CD4 T cells that are affected by the TGF-β inhibitor as this would guide the development of therapeutic strategies to target specific tumor-promoting CD4^+^ T cells in pancreatic tumors. 

Clinical studies with galunisertib (LY2157299 monohydrate) have demonstrated its safety and potential antitumor activity [55,56,57]. It is currently under clinical development in combination with checkpoint inhibitors in patients with non-small cell lung carcinoma (NSCLC), hepatocellular carcinoma (HCC) (NCT02423343), or pancreatic cancer (NCT02734160). In addition, there is an ongoing phase I/II study of galunisertib in combination with the anti-PD-1 antibody nivolumab in participants with advanced refractory solid tumors and in recurrent or refractory non-small cell lung cancer or hepatocellular carcinoma (metastatic and/or unresectable; NCT02423343). Anti-PDL1 mAb therapy is very effective but not all patients respond to this as single agent. The objective response rate with approved anti-PD-L1 mAb as monotherapy is ~20% in urothelial carcinomas [58,59,60], ~15% in non-small-cell lung cancer (NSCLC) [61,62], and ~30% in Merkel cell carcinoma [5,6]. Targeting TGF-β pathway inhibition represents an attractive strategy to enhance immune checkpoint blockade. Indeed, a recent study has shown that lack of response to atezolizumab (anti-PD-L1 mAb) in metastatic urothelial cancer patients was associated with active TGF-β signaling in peritumoral stroma and especially in patients with T cells excluded from the tumor parenchyma [27]. However, it is unclear whether lack of response to PD-L1 checkpoint blockade is also correlated with active TGF-β signaling in other patients of different tumor types. Furthermore, even though the combination of TGF-β blockade and checkpoint inhibitors has been demonstrated in multiple preclinical studies, their therapeutic efficacy varies across a range of syngeneic tumors [27,45,46,63,64,65]. Together, our studies indicate that adequate immune phenotyping of the various tumor models is critical for both rational model selection and data interpretation. This is critical as TGF-β has diverse and profound effects on the immune system, and therefore knowledge of the mechanisms by which TGF-β interferes in different tumor models may improve the current TGF-β-based immunotherapeutic approaches for specific tumor types.

## Figures and Tables

**Figure 1 cells-08-00320-f001:**
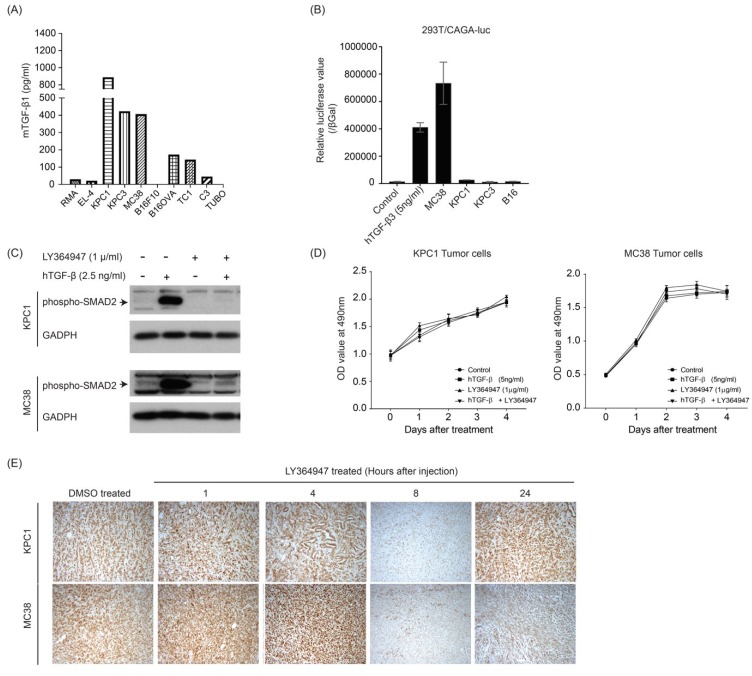
Production level of transforming growth factor-β (TGF-β) by various preclinical mouse tumor models and the potency of LY364947 to inhibit TGF-β-mediated cellular SMAD2 phosphorylation. Latent (**A**) and active (**B**) TGF-β in the conditioned media of cancer cell lines was assessed by TGF-β1 ELISA and transcriptional CAGA-luciferase reporter assay, respectively. (**C**) Immunoblotting of phospho-Smad2 of KPC1 and MC38 tumor cell lines after TGF-β and/or LY364947 treatment. glyceraldehyde 3-phosphate dehydrogenase (GAPDH) was measured as loading control. (**D**) Effect of the TGF-β and/or LY364947 on the proliferation of KPC1 and MC38 tumor cell lines. (**E**) Established MC38 or KPC1 tumor-bearing C57Bl/6 mice were administered LY364947 or DMSO, respectively. At 1, 4, 8, and 24 h after the injection, mice were sacrificed and tumors were analyzed by immune-histochemical staining for phospho-SMAD2.

**Figure 2 cells-08-00320-f002:**
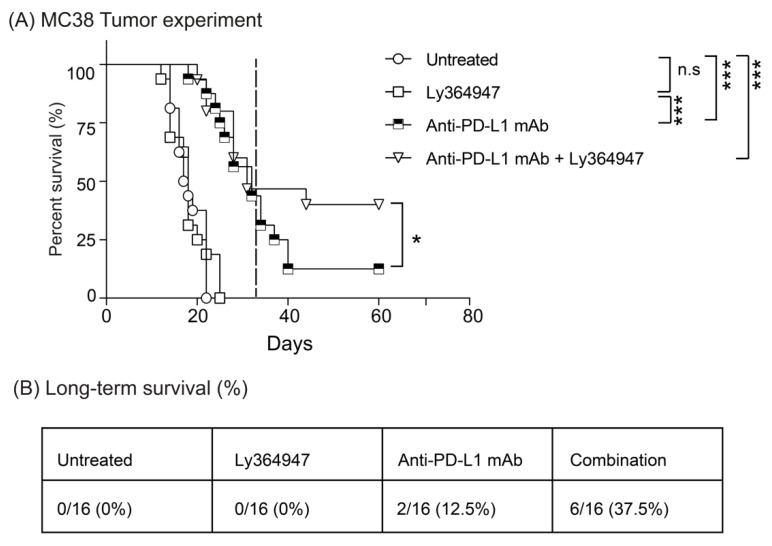
LY364947 improves anti-PDL1 mAb therapy. (**A**) MC38 tumor-bearing mice were treated with 200 μg anti-PDL1 mAb i.p. (MIH5; days 8, 10, and 13) and/or 10 mg/kg TGF-β receptor kinase inhibitor i.p. (LY364947; days 8, 10, 13, and every three days). Data presented as Kaplan–Meier survival curves with a total of 16 animals per group. Dashed line represents day 31. The log-rank test was used to determine the statistical significance of the survival. (**B**) Percentage of mice bearing subcutaneous MC38 treated with indicated regimens that rejected the tumor and survived tumor-free-long-term. Data compiled from two independent experiments, 16 mice per group. PDL1: programmed death-ligand 1. (* *p* <0.05; *** *p* <0.001, n.s, non signifiant).

**Figure 3 cells-08-00320-f003:**
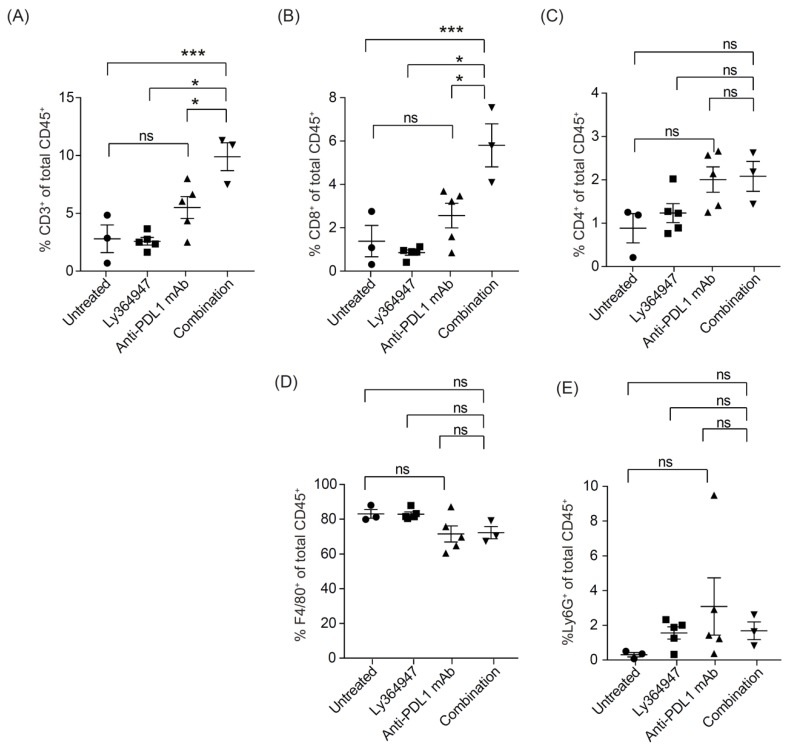
Combining TGF-β receptor kinase inhibitor LY364947 with anti-PDL1 mAb modulates infiltration of T cells in the tumor. Mice with established MC38 were treated with anti-PD-L1 mAb (200 μg; days 8, 10, and 13) and/or LY364947 (10 mg/kg; days 8, 10, 13, and 14). Tumors were harvested at day 15 and analyzed for the percentages of CD3 (**A**), CD8 (**B**), CD4 (**C**) T lymphocytes, (**D**) F4/80^+^ macrophages, and Ly6G^+^ granulocytes (**E**) by flow cytometry. Bars represent mean ± SEM. Statistical significance was determined by one-way ANOVA (* *p* < 0.05; *** *p* < 0.001; ns, non significant).

**Figure 4 cells-08-00320-f004:**
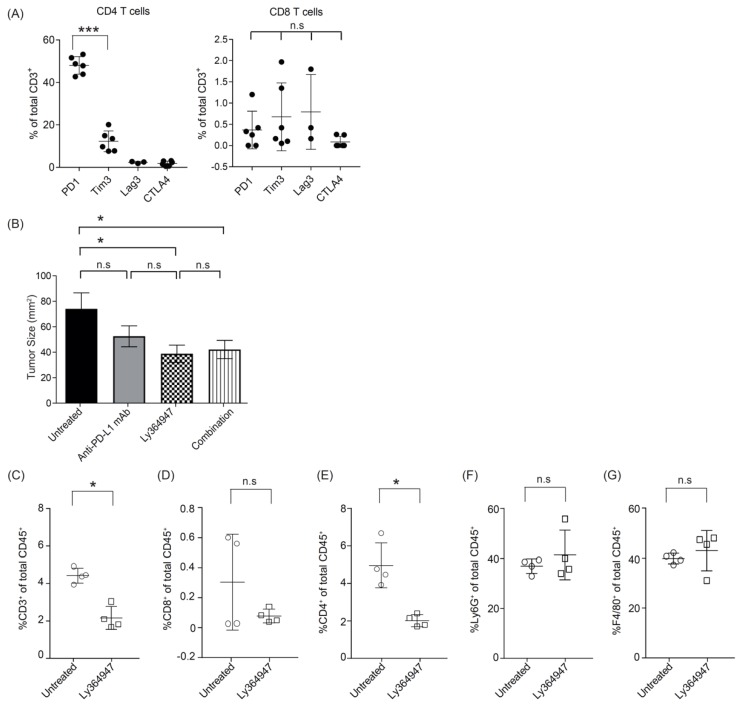
Antitumor effect of LY364947 in KPC1 tumor model. (**A**) Established KPC1 tumors were harvested at day 17 and analyzed for the percentages of PD-1^+^; Tim3^+^; Lag3^+^; CTLA4^+^ CD4^+^ or CD8^+^ T cells. (**B**) KPC1-tumor bearing mice were treated with 200 μg anti-PDL1 mAb (MIH5; days 8, 10, and 13) and/or 10 mg/kg TGF-β inhibitor (LY364947; day 8 and once every day). Data are represented as mean of tumor size mm^2^ ± SEM at day 23. Statistical significance was determined by two-way ANOVA (* *p* <0.05; *** *p* <0.001; n.s., non-significant). Data from one experiment, eight mice per group. (**B**) Mice with established KPC1 were treated with LY364947 (10 mg/kg; day 10 to 15). Tumors were harvested at day 16 and analysed for the percentages of (**C**) CD3^+^, (**D**) CD8^+^, (**E**) CD4^+^ T lymphocytes, (**F**) Ly6G^+^ granulocytes, and (**G**) F4/80^+^ macrophages. Statistical significance was determined by Student’s *t*-test (* *p* < 0.05; n.s, non significant).

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
