# Peer review of "Combined Inhibition of TGF-β Signaling and the PD-L1 Immune Checkpoint Is Differentially Effective in Tumor Models"

_cells, 2019, doi:10.3390/cells8040320_

Reviewer 1 Report

Recently, co-targeting TGFbeta signaling and blocking PD-L1 have been reported in several cancers. The authors described their test of co-targeting in MC38 and KPC1 tumor models. They mainly focused on reporting the outcomes of this co-targeting. Minimum mechanistic studies on the effect of T cells were performed. The study was well-designed and the manuscript was well written. Minor editing is still required though.

The major concern for this manuscript is the authors claim for “synergistic effect”. I suggest the authors either tone down this conclusion, or do the calculation to find out if there is indeed “Synergistic effect” exist.  

Author Response

Point-by-point reply

Reviewer 1

Comments to Authors

The major concern for this manuscript is the authors claim for “synergistic effect”. I suggest the authors either tone down this conclusion or do the calculation to find out if there is indeed “Synergistic effect” exist.  

We thank the reviewer for the suggestion and have toned down this conclusion accordingly:

LY364947 improves anti-PD-L1 mAb therapy (and removed “synergistic effect of these two regimens”)

Reviewer 2 Report

The authors have presented a somewhat compelling study regarding single and dual inhibition in cell culture and mice that targets TGFb signals and/or PD-1. This work is done in colon and prancreatic cancers in a similar format, allowing for a more direct comparison which is delivered in this manuscript. The work is nicely introduced and described, though some data points need some additional work/rigor to strengthen the overall study. There are a few major weaknesses including less than convincing significance in overall survival data, at least as presented and how well this particular model with low mutational status best mimics pancreatic cancer. There are a few other minor considerations as highlighted in the attached file. Despite these limitations, there is reason to believe that this study with some additional yet modest improvements can be impactful in the research community. 

Author Response

Point-by-point reply

Reviewer 2

Comments to Authors

TGF- is well known (line 48)

We apologize for this mistake and have added “β”.

Into tumor microenvironment (line 60)

We thank the reviewer for the suggestion and have changed to “into the tumor microenvironment”.

In quality (line 121)

We apologize for this mistake and have changed to in quantity.

Consider procuring protein lysate from PDAC from KPC mice to confirm/extend these findings. Likewise, a similar exercise for the colon cancer model. In addition, please include data here from blood collected from the orthotopic models used in subsequent findings below (line 158).

We thank the reviewer for the suggestion. However, we used subcutaneous models in our study, and we did not collect (blood) material for Western blot analysis

With at least a 2-fold increase in TGFb1 in KPC1 vs. MC38, this difference alone might compensate for the ability of LY to sufficiently block this signal for a long duration, like that observed in MC38 cells. Please consider this and perhaps a higher dose of LY to better block TGFb signals (line 173).

We thank the reviewer for suggestion. We agree with reviewer that higher dose of LY may better block TGF-b signal, in fact some galunisertib was injected at relatively high amount (from 75mg/kg to 800mg/kg). However, we are not sure how well the mice tolerated the high doses in those studies. In our study, we decided to go for 10mg/kg, this was due to the reported higher mortality rate when mice received 20mg/kg of TGF-b small molecule inhibitor (Antimicrob Agents Chemother. 2009 Nov; 53(11): 4694–4701). In order to bring this forward, a detailed study of the injection routes, dosing of TGF-b inhibitor and its toxicity profile should be first carried out. In our study, higher dose is difficult to achieve (for ip injection) because of limited solubility of LY compound in water, and more frequent dosing is not allowed by ethics committee.

Consider using dashed and dotted lines as well as different symbols (triangles up or down are difficult to decipher). A and B are basically the same. Consider keeping these as one larger graph (line 197).

We thank the reviewer for the suggestions. We have removed graph B and focus on graph A instead. On graph A, we have changed the symbol for combination group and added a dashed line represents Day 31.

It is highly suggested that representative quadrant capture plots of flow cytometry be used to show representative findings from this data set (line 218).

We thank the reviewer for the suggestion. We have now prepared a supplementary figure (Fig. S1) with representative plots showing the frequency of CD8 T cells (untreated, LY364947, anti-PD-L1 mAb and combination) in MC38 tumor and CD4 T cells (untreated and LY364947) in KPC1 tumor.

Is a low mutational burden commonly observed in human pancreatic cancer? Could this be something seen in the poorly immunogenic subtype of PC? In general, it seems that PC has a relatively high mutational burden as observed with 24-40 mutation events in a small cohort (Int J Oncol. 2018 Jun;52(6):1972-1980) and confirmed in a larger set (Gastroenterology. 2017 Jan;152(1):68-74). So, the model perhaps is limited in scope as it applies to the focus of this work. Perhaps use of a PC tumor cell line that has a high mutational burden would help to resolve this issue. (line 228)

We thank the reviewer for the question. According to the study by Alexandrov et al (Nature. 2013 Aug 22;500(7463):415-21), it was found through genomic analysis that PC had an average of 61 coding mutations while melanomas had an average of 511 coding mutations. KPC1 cell line is derived from a spontaneous pancreatic tumor of mutant KrasG12D; Trp53R172H, Pdx-1Cre (KPC) mice (Cancer Cell. 2005 May;7(5):469-83). Mutations in TP53 tumor suppressor gene and KRAS proto-oncogene are the two of the most commonly mutated genes in pancreatic cancer patients. Moreover, similar to the many human PC, KPC tumor exhibits relatively low mutational burden (JCI Insight. 2016 Sep 8; 1(14): e88328). In fact, KPC mouse model reproduces many of the key features of the immune microenvironment observed in human PDAC including a robust inflammatory reaction and exclusion of effector T cells. For this reason, we believe the KPC pancreatic model represents valuable model with a high potential of translational relevance for examining therapeutic efficacy of anti-PD-L1 mAb and LY364947 in our study. However, we agree with reviewer that some PC might have higher mutational burden, and this may have an impact on the therapeutic efficacy of anti-PD-L1 mAb and LY364947. Study with a PC tumor cell line such as Pan02 (derived from PDAC tumor induced by implanting 3-methyl-cholanthrene in the pancreas of C57Bl/6 mice) that has a higher mutational burden may help address this question. In the revised version of the manuscript, we have extended our discuss regarding this tumor model

Since FoxP3 is a marker of Tregs and TGFb maintains Tregs, one would think that suppressed TGFb would reduce FoxP3+ cells systemically. How do you account for this? (line 239)

In our study, the numbers of Tregs are not strongly affected by LY364947. This is in agreement with study from Mariathasan et al 2018 (Nature. 2018 Feb 22; 554(7693): 544–548.). Another recent study demonstrates that anti-CD25 mAb but not TGF-b inhibitor reduces the number of Tregs. Thus, it seems to suggest that TGF-b inhibitor has limited Treg depleting effect, but we cannot rule out the possibility that the inhibitor may reduce the suppressive immune responses induced by Tregs.

This data does not appear significant. Is the variance standard error of the mean? Not sure that two-way anova is best here. Overall, this data set is not convincing (line 246).

We apologize for this mistake and have changed the standard deviation to variance standard error of the mean. We have also analysed the data with Mann-Whitney test and found significant difference in between untreated and LY364947 or combination therapy treated groups.

 In consistent (line 259)

We apologize for this mistake and have changed to “is consistent”.

More favourable (line 270)

We apologize for this mistake and have added to “a” to more favourable.

Principle (line 279)

We apologize for this mistake and have changed to “principe”.

Investigates (line 291)

Investigates has been changed to “investigations”.

How do these findings compare to the ones generated in this report (line 312)?

We thank the reviewer for the question. Unlike most studies which illustrated the combination therapy in tumor model of similar immunogenicity, we compared the therapeutic efficacy of this regimen in immunogenic MC38 and non-immunogenic KPC1 tumor models. The anti-tumor activity of the combination therapy and the involvement of cytotoxic T cells in MC38 tumor model is consistent with the findings of multiple recent studies using immunogenic tumor models (CT26 colon cancer, EMT6 breast cancer, 4T1 breast cancer) which demonstrated the improvement of anti-PD-L1 mAb when it is combined TGF-β receptor kinase inhibitor anti-TGF-β mAb or galunisertib. Although combination of anti-PD-1 mAb and galunisertib has shown improvement in pancreatic KC tumor model, we are less convinced because the corresponding groups that received anti-PD-1 or Galunisertib monotherapy is missing in the Kaplan Meier curves (Mol Cancer Ther. 2019 Mar;18(3):613-620). We did not observe additive effect of combination therapy KPC tumor model, but our study revealed that blockade of TGF-β can delay the KPC tumor outgrowth by decreasing the relative amount of CD4 T cells within the tumor microenvironment suggesting that targeting specific tumor promoting CD4 T cells would help develop therapeutic strategy for pancreatic cancer. Taken together, out study highlights the need for selecting appropriate models for preclinical immunotherapy research, a greater understanding of model biology allows more robust interpretation between response and mechanism of action of therapeutic agents that may ultimately help improve the efficiency of drug discovery.

Reviewer 3 Report

The study reported in the manuscript entitled "Combined inhibition of TGF-β signaling and the PD-L1 immune checkpoint is differentially effective 3 in tumor models " is very interesting and well structured.

The results are clearly described and well commented.

The illustrated data could provide important information on the importance of correctly establishing combined therapeutic strategies in different tumor types.

Author Response

Point-by-point reply

Reviewer 3

The study reported in the manuscript entitled "Combined inhibition of TGF-β signaling and the PD-L1 immune checkpoint is differentially effective 3 in tumor models " is very interesting and well structured.

The results are clearly described and well commented.

The illustrated data could provide important information on the importance of correctly establishing combined therapeutic strategies in different tumor types.

We thank the reviewer for reviewing our manuscript.